# Prior knowledge meets Neural ODEs: a two-stage training method for improved explainability

**C. Coelho**
Centre of Mathematics (CMAT)
University of Minho
`cmartins@cmat.uminho.pt`

**M. Fernanda P. Costa**
Centre of Mathematics (CMAT)
University of Minho
`mfc@math.uminho.pt`

**L.L. Ferrás**
Department of Mechanical Engineering - Section of Mathematics
FEUP - University of Porto
Centre of Mathematics (CMAT)
University of Minho
`lferras@fe.up.pt`

## Abstract

Neural Ordinary Differential Equations (ODEs) have been used extensively to model physical systems because they represent a continuous-time function that can make predictions over the entire time domain. However, most of the time, the parameters of these physical systems are subject to strict laws/constraints. But there is no guarantee that the Neural ODE model satisfies these constraints. Therefore, we propose a two-stage training for Neural ODE. The first stage aims at finding feasible parameters by minimizing a loss function defined by the constraints violation. The second stage aims at improving the feasible solution by minimizing the distance between the predicted and ground-truth values. By training the Neural ODE in two stages, we ensure that the governing laws of the system are satisfied and the model fits the data.

## 1 Introduction

Neural networks (NN) have proven useful in modeling physical, biological, and chemical systems (Kutz, 2017; Soleymani et al., 2022; Atz et al., 2021). Recently, Chen et al. (2018) proposed Neural ODEs, a NN that models a continuous-time function, ODE, to the hidden dynamics of the data. Because of their ability to handle irregularly sampled data and make predictions throughout the time domain, Neural ODEs have been used to model natural systems (Xing et al., 2022; Su et al., 2022). In general, natural systems follow strict rules that are known. Therefore, a NN modeling them would ideally operate strictly according to the governing laws. However, due to the black-box approach, there is no guarantee that these constraints will be extracted from the data (Greydanus et al., 2019). The lack of understanding of the laws governing natural systems can lead to inconclusive predictions, which raises the distrust of experts in the field (Dobbelaere et al., 2021). A popular way to ensure that these constraints are satisfied is to add penalty terms to the loss function (Nocedal & Wright, 2006). This approach has already been used in the context of Neural ODEs (Tuor et al., 2020). However, when combining the original loss function $l(\boldsymbol{\theta})$, which measures the agreement between the prediction and the ground truth, with a measure of the constraint violations, $c(\boldsymbol{\theta})$, a positive constant $\mu$ (penalty hyperparameter) must be given to penalize the constraint violation. Unfortunately, setting a suitable $\mu$ is still a problematic issue, which requires a priori experimentation (Nocedal & Wright, 2006). Therefore, we introduce a new two-stage training Neural ODE technique to avoid the usage of penalty terms while satisfying the constraints.

## 2 METHOD

We propose a two-stage training method to incorporate prior knowledge on the constraints of the system into Neural ODEs without using penalty terms. The method is composed of two stages: admissibility phase; and optimization phase. In phase I the goal is to find an admissible solution by training the Neural ODE with a violation loss function $\mathcal{L}_I(\boldsymbol{\theta}) = \sum_{i \in \Gamma} |c_i^t(\boldsymbol{\theta})| + \sum_{j \in \Upsilon} \|max(0, c_j^t(\boldsymbol{\theta}))\|$, where $c_i^t(\boldsymbol{\theta}) = 0, i \in \Gamma$ and $c_j^t(\boldsymbol{\theta}) \geq 0, j \in \Upsilon$ are the prior knowledge constraints, with $t = t_1, \ldots, t_f$. Phase I training ends when the loss value is below a given threshold.

The parameters at the end of phase I are the starting point for phase II. In phase II, the loss function is given by $l(\theta)$, $\mathcal{L}_{II}(\boldsymbol{\theta}) = l(\boldsymbol{\theta})$, which measures the agreement between the prediction and the ground truth. To keep the search from leaving the admissibility region during phase II, two different strategies are used. If the violation loss function $\mathcal{L}_I$ at the new current solution yields a higher constraint violation value, the point solution is discarded. In this case, either the parameters from the previous iteration are kept (*updatePrevious*) or the best iteration parameters are used instead (*updateBest*).

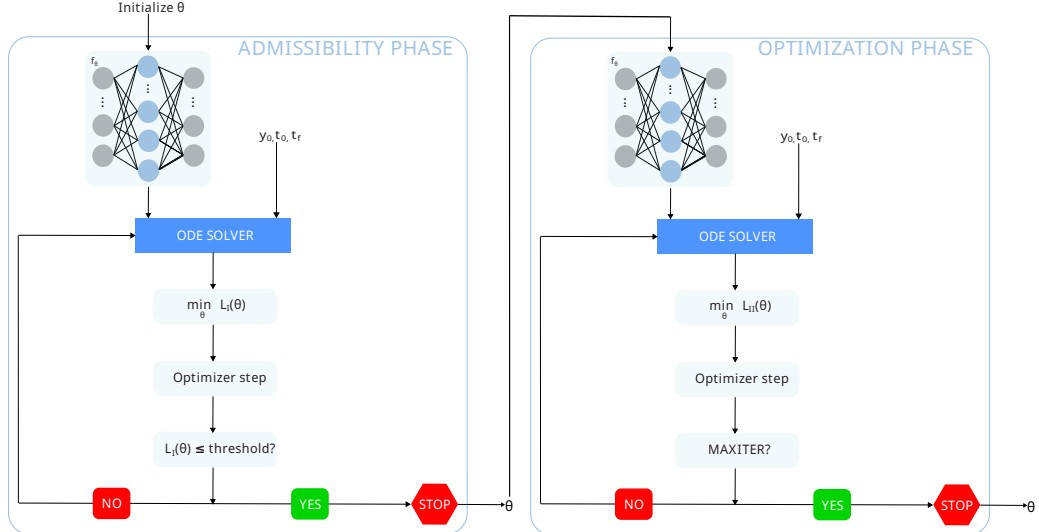

Figure 1: Schematic of the two-stage training method for Neural ODE with initial condition $y(t_0) = y_0$ and solving in the time interval $(t_0, t_f)$.

The proposed method was evaluated numerically by modelling the growth of the world population with an upper boundary constraint defined by the carrying capacity, and a chemical reaction with an equality constraint for the conservation of mass. The experimental details and results for these natural systems can be found in A.1 and A.2, respectively.

## 3 CONCLUSION

The proposed two-stage training method successfully introduces prior knowledge on the physical system into the Neural ODE by avoiding penalty hyperparameters. By using two separate training phases, we ensure that the constraints are satisfied while the agreement between the predictions and the ground-truth is minimized. The explicit introduction of constraints contributes to the explainability of the Neural ODE models. Moreover, numerical experiments demonstrate the robustness of the models produced with our method. In the future, further investigation will be carried out to facilitate the choice between *updatePrevious* and *updateBest*. In addition, numerical experiments will be conducted to model more complex natural systems (with more constraints).

ACKNOWLEDGEMENTS

We would like to thank the referees for their comments, which helped improve this paper considerably. The authors acknowledge the funding by Fundação para a Ciência e Tecnologia (Portuguese Foundation for Science and Technology) through CMAT projects UIDB/00013/2020 and UIDP/00013/2020. C. Coelho would like to thank FCT for the funding through the scholarship with reference 2021.05201.BD.

URM STATEMENT

All authors meet the URM criteria of ICLR 2023 Tiny Papers Track.

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

## A    NUMERICAL EXPERIMENTS

For each case study (world population growth (Coelho et al., 2023b) and chemical reaction evolution (Coelho et al., 2023a)), three different experiments were conducted: same data for training/testing (reconstruction); larger test time horizon (extrapolation); same time interval for training and testing but higher frequency sampling in the testing set, *i.e.*, using a smaller step size (completion). Performance was analysed by Mean Squared Error (MSE), sum of constraint violations and standard deviation (std).

### A.1    POPULATION GROWTH

The World Population Growth Dataset was synthesized and has one feature, world population $p(t)$. The goal is to predict the population at each point in time. The system has an inequality constraint defined by the carrying capacity $p(t) \leq 12$ at each time step $t$ with $t = t_1, \ldots, t_N$. For the reconstruction task, the training and testing datasets are the same. For the extrapolation task, the training was done with 200 data points in the time interval $[0, 300]$ and 200 test points in the interval $[0, 400]$. For the completion task, 200 training and 300 testing points are used in the interval $[0, 300]$ (Coelho et al., 2023b).

The NN has 4 hidden layers: linear with 50 neurons; hyperbolic tangent (tanh); linear with 50 neurons; Exponential Linear Unit (ELU). The input and output layers have 1 neuron. The Adam optimizer was used with a learning rate of $1e - 5$, an admissibility threshold of $1e - 4$, and 10000 iterations.

The results of each experiment are shown in the 1 and 2 tables for the strategies *updatePrevious* and *updateBest*, respectively. In addition, the ground-truth and predicted values for the testing dataset were plotted for the strategies *updatePrevious*, Figure 2, and *updateBest*, Figure 3.

Table 1: Performance on the population growth testing dataset with *updatePrevious* strategy(N=3).

| EXPERIMENT | MSE±STD | VIOLATION±STD |
|---|---|---|
| Reconstruction | $0.007954 \pm 0.004054$ | $0.009379 \pm 0.004777$ |
| Extrapolation | $0.013587 \pm 0.013542$ | $0.035037 \pm 0.030044$ |
| Completion | $0.008397 \pm 0.004406$ | $0.011079 \pm 0.005657$ |

Table 2: Performance on the population growth testing dataset with *updateBest* strategy (N=3).

| EXPERIMENT | MSE±STD | VIOLATION±STD |
|---|---|---|
| Reconstruction | $0.010587 \pm 0.002240$ | $0.013616 \pm 0.001424$ |
| Extrapolation | $0.030280 \pm 0.012093$ | $0.069097 \pm 0.012446$ |
| Completion | $0.012709 \pm 0.006797$ | $0.013774 \pm 0.005567$ |

### A.2    CHEMICAL REACTION

The chemical reaction dataset is a synthetic reaction with four chemical components defined by $A + B \rightarrow C + D$. The goal is to predict the mass of each component over time. At each time step $t$, this system must satisfy an equality constraint defined by the conservation of mass: $A(t) + B(t) + C(t) + D(t) = m_{\text{total}}$, with $m_{\text{total}}$ the total constant mass of the system. For the reconstruction task, the training and testing datasets are identical. For the extrapolation task, training was performed with 100 data points in the time interval $[0, 100]$ and 100 test points in the interval $[0, 200]$. For the completion task, 100 training and 200 testing points are used in the interval $[0, 100]$ (Coelho et al., 2023a).

The NN has 6 hidden layers: linear with 50 neurons; tanh; linear with 64 neurons; ELU; linear with 50 neurons; tanh. The input and output layers have 4 neurons. The Adam optimizer was used with a

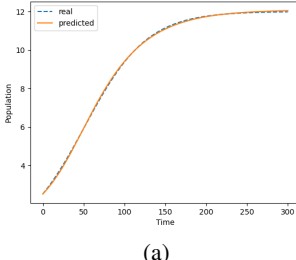 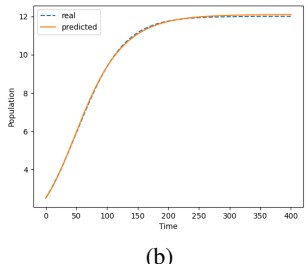 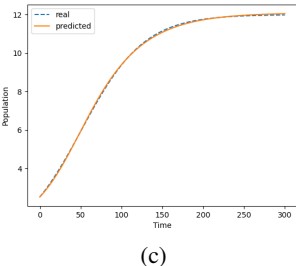

(a)          (b)          (c)

Figure 2: Plot of the real (dashed line) and predicted (solid line) curves, for the world population growth, on different tasks using *updatePrevious*: (a) Reconstruction; (b) Extrapolation; (c) Completion.

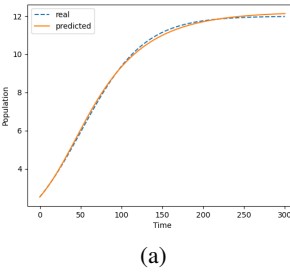 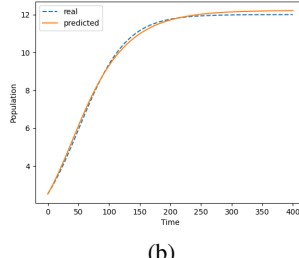 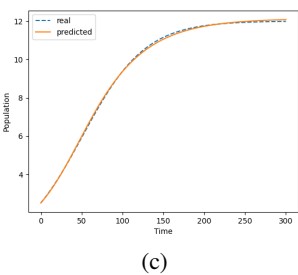

(a)          (b)          (c)

Figure 3: Plot of the real (dashed line) and predicted (solid line) curves, for the world population growth, on different tasks using *updateBest*: (a) Reconstruction; (b) Extrapolation; (c) Completion.

learning rate of $1e-5$, an admissibility threshold of $1e-4$, and $10000$ iterations. In this experiment, the NN architecture used is more complex given that the problem is harder.

The results of each experiment are shown in the 3 and 4 tables for the strategies *updatePrevious* and *updateBest*, respectively. Letters A to D represent masses of the chemical species in the reaction, real (dashed line) and predicted (solid line) values for the testing dataset were plotted for the strategies *updatePrevious*, Figure 4, and *updateBest*, Figure 5.

Table 3: Performance on the chemical reaction testing dataset with *updatePrevious* strategy (N=3).

| EXPERIMENT | MSE$\pm$STD | VIOLATION$\pm$STD |
|---|---|---|
| Reconstruction | $0.000104 \pm 5.26688e-5$ | $0.006874 \pm 0.003609$ |
| Extrapolation | $9.66667e-5 \pm 5.92021e-5$ | $0.091390 \pm 0.073354$ |
| Completion | $0.000181 \pm 0.000156$ | $0.030485 \pm 0.012531$ |

Table 4: Performance on the chemical reaction testing dataset with *updateBest* strategy(N=3).

| EXPERIMENT | MSE±STD | VIOLATION±STD |
|---|---|---|
| Reconstruction | $0.000461 \pm 0.000536$ | $0.113920 \pm 0.058010$ |
| Extrapolation | $0.001244 \pm 0.001665$ | $0.143724 \pm 0.108796$ |
| Completion | $0.001007 \pm 0.000914$ | $0.027309 \pm 0.013076$ |

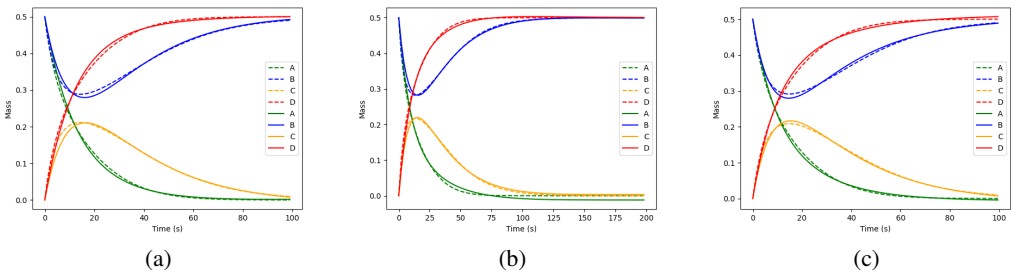

      (a)                (b)                (c)

Figure 4: Plot of the real (dashed lines) and predicted (solid lines) curves, for the chemical reaction, on different tasks using *updatePrevious*: (a) Reconstruction; (b) Extrapolation; (c) Completion.

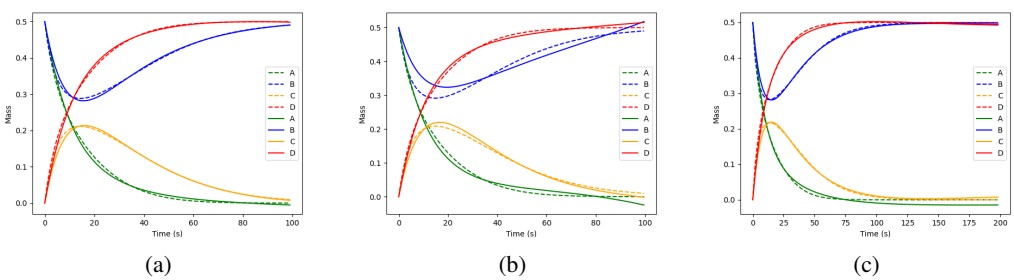

      (a)                (b)                (c)

Figure 5: Plot of the real (dashed lines) and predicted (solid lines) curves, for the chemical reaction, on different tasks using *updateBest*: (a) Reconstruction; (b) Extrapolation; (c) Completion.

