# OpenReview forum: "Prior knowledge meets Neural ODEs: a two-stage training method for improved explainability"
_ICLR.cc/2023/TinyPapers — Submitted to Tiny Papers @ ICLR 2023_

### Official Review · Reviewer_FU6k · 2023-04-02

**Confidence:** 3

**Summary Of Contributions:**

The authors propose training neural ODEs in two-stages. At stage 1 the model is optimized to respect the constraints of the system being modeled. At stage 2 the model is further optimized to approximate the ground truth. Importantly, intermediate solutions that violate the constraints are rejected, guiding the optimization process towards a final solution that delivers both a low prediction error and respects the constraints (up to some tolerance).

**Rating:**

Clear, Correct, and Reproducible (CCR): a submission which meets the reviewing criteria

**Strengths And Weaknesses:**

Strengths: Overall, the paper is well motivated, addresses an interesting problem and is written clearly. The method presents an interesting way to avoid having to choose the weight for the constraint violation penalty.

Weaknesses:
It is not clear at all from the Method section what constraints were imposed on the model.
Specifically, if $\Gamma$ and $\Upsilon$ are two “prior knowledge constraints”, then why do they depend on $\theta$? If that is intended, then that should be made clear. An explanation and/or references are needed to help the reader understand this essential part of the paper.

There is not information about the nature of the constraints for the Population Growth case (Appendix 1) and Chemical Reaction (Appendix 2).

The difference between “completion” and “extrapolation” (first paragraph of the Appendix) is not clear and should be defined.

The “World Population Growth” and “chemical reaction” datasets should be either briefly explained or linked to their source.


**Suggested Changes:**

I suggest the authors make the following changes:

1) Add minimal information about the constraints used in the Population Growth case (Appendix 1) and Chemical Reaction (Appendix 2).

2) Explain why $\Gamma$ and $\Upsilon$ are indexed with different indices and why they are a function of $\theta$.

3) Clarify the difference between “completion” and “extrapolation” (first paragraph of the Appendix).

4) Provide explanation about (or reference to) the “World Population Growth” and “chemical reaction”.

---

### Official Review · Reviewer_KtNV · 2023-04-02

**Confidence:** 2

**Summary Of Contributions:**

The paper introduces a new two-stage method for solving Neural ODE optimisation problems with additional parameter constraints. Instead of incorporating constraint-related terms in the optimisation loss of the Neural ODE, the paper proposes to iterate between two optimisation sub-problems, where one trains the Neural ODE to comply with the constraints, and the second performs unconstrained optimisation of the Neural ODE with a loss that improves predictions into the future.

**Rating:**

Clear, Correct, and Reproducible (CCR): a submission which meets the reviewing criteria

**Strengths And Weaknesses:**

++
+ Well explained motivated problem setting.
+ Well formulated proposed approach.

--
- As a non familiar reader with constrained optimisation of NODEs, I did not get convinced of the advantage of this approach over the usual single-stage extended loss approach or some variant of the usual approach.

- The sampling of the points seems rather dense to me given the timescales of the observed systems.


**Suggested Changes:**

- If possible, show a small comparison where the extended loss single-stage optimisation fails or is slow to converge, while the presented method is successful/more efficient.
- Also an ablation experiment for the first example you present (which is quite simple and should be trivial to perform as I understand), without the admissibility stage would help to convince the reader that this stage is indeed necessary, or otherwise the predicted population size would have exceeded the upper bound.
- Formatting suggestion: Increase the font size of the numbers/labels of your plots.
- Possibly for future work: How does the performance of  your approach scale with the density of the training points (bigger inter-observation distances)?
- Please provide the associated implementation for the experiments, because otherwise it would be difficult to reproduce the results from the descriptions in the appendix.

---

### Official Review · Reviewer_Cu1K · 2023-04-03

**Confidence:** 3

**Summary Of Contributions:**

Prior Neural ODEs required a ‘constraints’ penalty term in the loss function to help the NN extract the system’s physical constraints, latent in the training dataset. This penalty term is scaled by a hyper-parameter that is difficult to determine. The authors discard the penalty term, learning system feasible parameters in an ‘admissibility’ stage, with the 2nd stage optimizing for agreement between prediction and ground truth data.

**Rating:**

High Impact (HI): a submission which meets the reviewing criteria and is predicted to make an impact on the field

**Strengths And Weaknesses:**

Strengths
- The limitations of prior method are well-motivated, providing a compelling argument for the need for this work.
- The method is communicated clearly, despite the conciseness.
- Figure 1 provides a nice overview of the method, communicating clearly most components.
- The results from the population growth and chemical reaction experiments show the breadth and potential impressive accuracy of this method.



Weaknesses
- Please see some suggested changes below. Otherwise, I think the main weakness of this work is that there are no comparisons to baselines. I realize the nature of the selection of $\mu$ may make these comparisons not apples to apples, but I believe this should be addressed.



**Suggested Changes:**

Clarity:
 - Not all the variables in Fig 1 are defined, forcing the reader to assume, e.g. $t_i$, $t_f$.
 - The described algorithm bares some loose resemblance to the Expectation-Maximization algorithm. Assuming there are relevant prior works with similarities to the method introduced here, it would be nice to cite them.
 - Nit:
   - It appears the ODE acronym is defined twice and with distinction. It may be preferable to use commas or restructure sentence:  "a NN that models a continuous-time function (ODE) to the hidden dynamics..."

Reproducibility:
 - Linking to source code would also greatly help reproducibility, but assuming that is not yet possible...
 - Explanatory details about stopping / converge criteria, or setting of `MAXITER` would be helpful.
 - Some model hyper-parameters vary for the population growth vs chemical reaction experiments. Some explanation about why different hyper-parameters where chosen and perhaps what did not work, could be helpful.

---

### Official Review · Reviewer_m5DW · 2023-04-03

**Confidence:** 4

**Summary Of Contributions:**

This paper proposed a two-stage training strategy for neural ordinary differential equation (ODE), which incorporates prior knowledges to satisfy the constraints and bypasses the penalty term in the existing method. This approach also improves the explainability of current neural ODE models.

**Rating:**

Great Start (GS): a submission which meets some of the reviewing criteria but has room for improvement

**Strengths And Weaknesses:**

**Strengths**

- This paper is well-motivated, i.e., satisfying constraints with incorporating prior knowledge while bypassing the usage of penalty terms in current neural ODE, which is an interesting research topic.
- The proposed method is reported to be effective.

**Weaknesses**

To provide more support for the claims, it would be helpful to include more thorough discussions of the experiments.

- "We ensure that the governing laws of the system are satisfied and the model fits the data."

  A clear and detailed discussion of which and how the laws are satisfied could be included. Baselines should be included to help readers to better interpret the numbers. This will help to mitigate the possibility that the dataset is simple enough for any models to fit. Two possible baselines:

  1. $\tt{cODE_G}$ as in "*Constrained neural ordinary differential equations with stability guarantee"*
  2. Proposed method without admissibility training phase.

- "The explicit introduction of constraints contributes to the explainability of the Neural ODE models."

  Perhaps more discussion around how explainability is improved with the proposed method would be useful.

- "Moreover, numerical experiments demonstrate the robustness of the models produced with our method."

  Is it robust to anomalous data points? Or lengthy time series? A clearer discussion in the current version would be useful.

- "setting a suitable $\mu$ is still a problematic issue"

  Is setting a suitable admissibility threshold a problematic issue? A discussion of the impact of the threshold is expected.

Generally, the method part in the current version is unclear. Here are several points I found confusing and need clarification:

- If $c_i(\mathbb{\theta})=0$ and $c_j(\mathbb{\theta})\geq0$ as defined in the violation loss function $\mathcal L_I$ are true, then can  $\mathcal L_I$ be simply formulated as $\sum_{j\in \Upsilon}\lVert{c_j(\theta)}\rVert$? I am not so sure if there are typos here.
- Besides the instantiation of the prior knowledge constraints $c_i(\mathbb{\theta})$ and $c_j(\mathbb{\theta})$  in the experiment part (e.g., carrying capacity), a formal definition or unified description of $c(\mathbf{\theta})$ is expected when introducing the loss function.
- Instead of terming it as "original loss function" or "agreement", a formal definition of $l(\mathbf{\theta})$ should be given.
- What do $\Upsilon$ and $\Gamma$ represent?
- Why there are two constraints of the violation loss ($c_i$, $c_j$) but in a later description it seems there is only one for each problem setting (carrying capacity, equality constraint)?
- What do $t_i$ and $t_f$ in figure 1 represent?
- Is std in table 1-3 for data points over time in one single run or for different random runs?

**Suggested Changes:**

The manuscript would be improved if the authors could address the following concerns in their future version.

- Is penalty term $\mu$ really unwanted? Is it worth introducing an additional training phase just to bypass this penalty?
- How much extra computation does the additional phase introduce?

Minor:
- A psuedo-algorithm of the proposed method might help with clarification.

- Missing references for both datasets. What is the dataset size?

- Experiments to support the claims should be included in the main text.

- Captions for figure 4-5 should indicate A, B, C, D are compounds in the chemical problem setting.

---

### Author Response · Authors · 2023-05-30
**wish to opt-in for archival**

We wish to opt-in for archival and adhere that all requirements are satisfied.

We want to thank all the reviewers for their comments.

---

> ### Comment · Area_Chair_PWuM · 2023-06-06
> **Check for archival**
>
> This work meets the threshold for archival, contents the URM statement and is deanonymized.

---

### Meta-Review · Area_Chair_PWuM · 2023-04-07

**Recommendation:** Invite to archive
**Confidence:** 4

**Metareview:**

This paper proposes a two-stage training strategy for neural ordinary differential equations (ODEs) that aims to incorporate prior knowledge constraints and achieve better explainability. Reviewers agree that this work is well-motivated for a compelling research problem, with figures and plots improving the reproducibility of this work. The experiment settings also ease reproducibility.

However, the reviewers suggest that the work lacks comparisons/ablation studies to further show the utility and advantage of the proposed training method for neural ODEs. Reviewer m5DW provides potential baselines for further comparisons.

In addition, the writing clarity of the method section could be further polished, including notions and claims. Adding a table of notions will be favorable to better understanding.

**Summary:**

The paper proposes a two-stage training strategy for neural ordinary differential equations (ODEs) that aims to incorporate prior knowledge constraints and achieve better explainability. The proposed method is demonstrated to be effective in reducing the violation of constraints.

**Reason For Not Giving A Higher Recommendation:**

Although this paper provides an interesting approach to addressing the research problem, there is still room for improvement. The lack of comparisons and ablation studies to further demonstrate the effectiveness of the proposed method is a significant limitation.

**Reason For Not Giving A Lower Recommendation:**

The proposed two-stage training strategy for neural ODEs is well-motivated, with detailed experiment settings that improve reproducibility. Although there are some weaknesses identified by the reviewers, the paper still meets the criteria of being Clear, Correct, and Reproducible (CCR).

---

### Decision · Program_Chairs · 2023-04-10

Invite to archive